# Different B cell subpopulations show distinct patterns in their IgH repertoire metrics

**Marie Ghraichy[1], Valentin von Niederhäusern[1], Aleksandr Kovaltsuk[2], Jacob D Galson[1,3], Charlotte M Deane[2], Johannes Trück[1]***

[1]Division of Immunology, University Children's Hospital and Children's Research Center, University of Zurich (UZH), Zurich, Switzerland; [2]Department of Statistics, University of Oxford, Oxford, United Kingdom; [3]Alchemab Therapeutics Ltd, London, United Kingdom

**Abstract** Several human B cell subpopulations are recognised in the peripheral blood, which play distinct roles in the humoral immune response. These cells undergo developmental and maturational changes involving VDJ recombination, somatic hypermutation and class switch recombination, altogether shaping their immunoglobulin heavy chain (IgH) repertoire. Here, we sequenced the IgH repertoire of naïve, marginal zone, switched and plasma cells from 10 healthy adults along with matched unsorted and *in silico* separated CD19+ bulk B cells. Using advanced bioinformatic analysis and machine learning, we show that sorted B cell subpopulations are characterised by distinct repertoire characteristics on both the individual sequence and the repertoire level. Sorted subpopulations shared similar repertoire characteristics with their corresponding *in silico* separated subsets. Furthermore, certain IgH repertoire characteristics correlated with the position of the constant region on the IgH locus. Overall, this study provides unprecedented insight over mechanisms of B cell repertoire control in peripherally circulating B cell subpopulations.

**\*For correspondence:**
Johannes.Trueck@kispi.uzh.ch

## Introduction

B cell development starts in the bone marrow where immature B cells must assemble and express on their surface a functional but non-self-reactive B cell antigen receptor (BCR) (*Lefranc and Lefranc, 2001*). The generation of the heavy and light chain of the BCR is mediated by the random and imprecise process of V(D)J recombination (*Tonegawa, 1983*). Further development of B cells occurs in the periphery in response to stimulation with the process of somatic hypermutation (SHM) through which point mutations are introduced in the genes coding for the V(D)J part of the immunoglobulin heavy (IgH) and light chain (*Jolly et al., 1996*). Subsequently, B cells with a mutated BCR providing increased antigen affinity are selected and show increased survival and proliferation capacity (*Zheng et al., 2005*).

Furthermore, class switch recombination (CSR) modifies the IgH constant region, resulting in the generation of B cells with nine different immunoglobulin isotypes or isotype subclasses, namely IgD, IgM, IgG1-4, IgA1/2 and IgE (*Stavnezer et al., 2008*). This process involves the replacement of the proximal heavy chain constant gene by a more distal gene. Class switching is an essential mechanism during humoral immune responses as the constant region of an antibody determines its effector function (*Vidarsson et al., 2014*). Both direct switching and sequential switching upon a second round of antigen exposure have been reported (*Horns et al., 2016*; *Cameron et al., 2003*; *Zhang et al., 1994*).

Through developmental mechanisms and further differentiation in the periphery, several phenotypically distinct circulating B cell subpopulations are generated (*Allman and Pillai, 2008*). They include naïve, marginal zone (MZ), switched memory B cells and plasma cells (PC), which are mainly characterised by their differential expression of surface markers and by playing distinct roles in the adaptive immune response (*Leandro, 2013*) High-throughput sequencing of the IgH repertoire (AIRR-seq) has made it possible to improve our understanding of the different components of the adaptive immune system in health and disease, and following vaccine challenge (*Mandric et al., 2020*; *Ghraichy et al., 2018*; *Galson et al., 2014*; *Lindau and Robins, 2017*; *Georgiou et al., 2014*). Previous studies using both high- and low-throughput sequencing techniques have already reported important differences between B cell subpopulations affecting their IgH repertoire composition, VDJ gene usage, mutations and clonality (*Berkowska et al., 2011*; *Mroczek et al., 2014*; *Galson et al., 2015*; *Wu et al., 2010*).

Recent AIRR-seq workflows allow coverage of a sufficient part of the IgH constant region in addition to the VDJ region, making it possible to assign antibody classes and subclasses on an individual sequence level. It is common practice to use unsorted bulk B cells from peripheral blood as a starting material and use the constant region information combined with the degree of SHM to group transcripts *in silico* into different B cell populations (*Glanville et al., 2011*; *Ghraichy et al., 2020*). Using isotype-resolved IgH sequencing of bulk B cells, isotype subclasses have been found to show differences in their repertoire characteristics (*Jackson et al., 2014*; *Kitaura et al., 2017*). However, it remains unknown how the IgH repertoire of bioinformatically separated transcripts originating from bulk-sequenced B cells compares to the repertoire of their corresponding circulating B cell subpopulations. It is also unknown how IgH sequences with the same constant region originating from different cell types compare.

Here, we used an established AIRR-seq workflow that captures the diversity of the variable IgH genes together with the isotype subclass usage to study in detail the repertoire of $CD19^+$ bulk B cells as well as flow cytometry sorted naïve, MZ, switched and PCs from 10 healthy adults. We applied statistical methods such as principal component analysis (PCA), k-means clustering, linear discriminant analysis (LDA) and machine learning algorithms to combine several repertoire metrics and characterise the different B cell subpopulations. We show that transcripts from physically sorted B cell subpopulations share similar characteristics with their corresponding subsets in the bulk that were grouped *in silico* using isotype subclass information and number of mutations. We further demonstrate that sequences with the same isotype subclass originating from different cell types are closely related, suggesting the presence of isotype-specific rather than cell-type-specific signatures in the IgH repertoire. We finally correlate these signatures to the isotype subclass positioning on the locus and find that downstream subclasses exhibit enhanced signs of maturity, overall providing new insights into the selection and the peripheral differentiation of distinct B cell subpopulations.

## Results

### Physically sorted B cell subpopulations and their corresponding subsets in the bulk share similar repertoire characteristics

We compared IgH repertoire characteristics between the following B cell subpopulations: $B_{naive}$, $B_{MZ}$, $B_{PC\_MD}$, $B_{PC\_AG}$ and $B_{switched}$ and their corresponding subsets that we obtained *in silico* from $B_{bulk}$: $B_{bulk\_naïve}$, $B_{bulk\_MD}$ and $B_{bulk\_switched}$. We identified three separate clusters: one made of predominantly $B_{MZ}$, $B_{bulk\_MD}$ and $B_{PC\_MD}$; another with only $B_{naive}$ and $B_{bulk\_naïve}$; and a third cluster with predominantly $B_{bulk\_switched}$, $B_{PC\_AG}$ and $B_{switched}$ (*Figure 1A*) by combining all repertoire characteristics in a PCA and applying k-means clustering. To test whether this clustering pattern was driven by VJ gene usage, complementarity-determining region (CDR) 3 physiochemical properties or the general repertoire metrics, we analysed these variables separately. Using V family and J gene usage, there was a clear separation between naïve and memory cells mostly driven by differences in V1/3 and J4/6 usage (*Figure 1—figure supplement 1*). However, no separation between $B_{MZ}$/$B_{PC\_MD}$/ $B_{bulk\_MD}$ and $B_{switched}$/$B_{PC\_AG}$/$B_{bulk\_switched}$ was observed (*Figure 1B*). The CDR3 physiochemical properties alone created similar clusters as when combined together with the other metrics (*Figure 1C*). This separation was mostly driven by a lower basic and a higher aromatic content in addition to a higher gravy index and a lower polarity in $B_{naïve}$/$B_{bulk\_naïve}$ compared to memory subpopulations (*Figure 1—figure supplement 2*). Global repertoire metrics also created a clear separation between $B_{naïve}$/$B_{bulk\_naïve}$, $B_{switched}$/$B_{PC\_AG}$/$B_{bulk\_switched}$ and $B_{MZ}$/$B_{PC\_MD}$/$B_{bulk\_MD}$

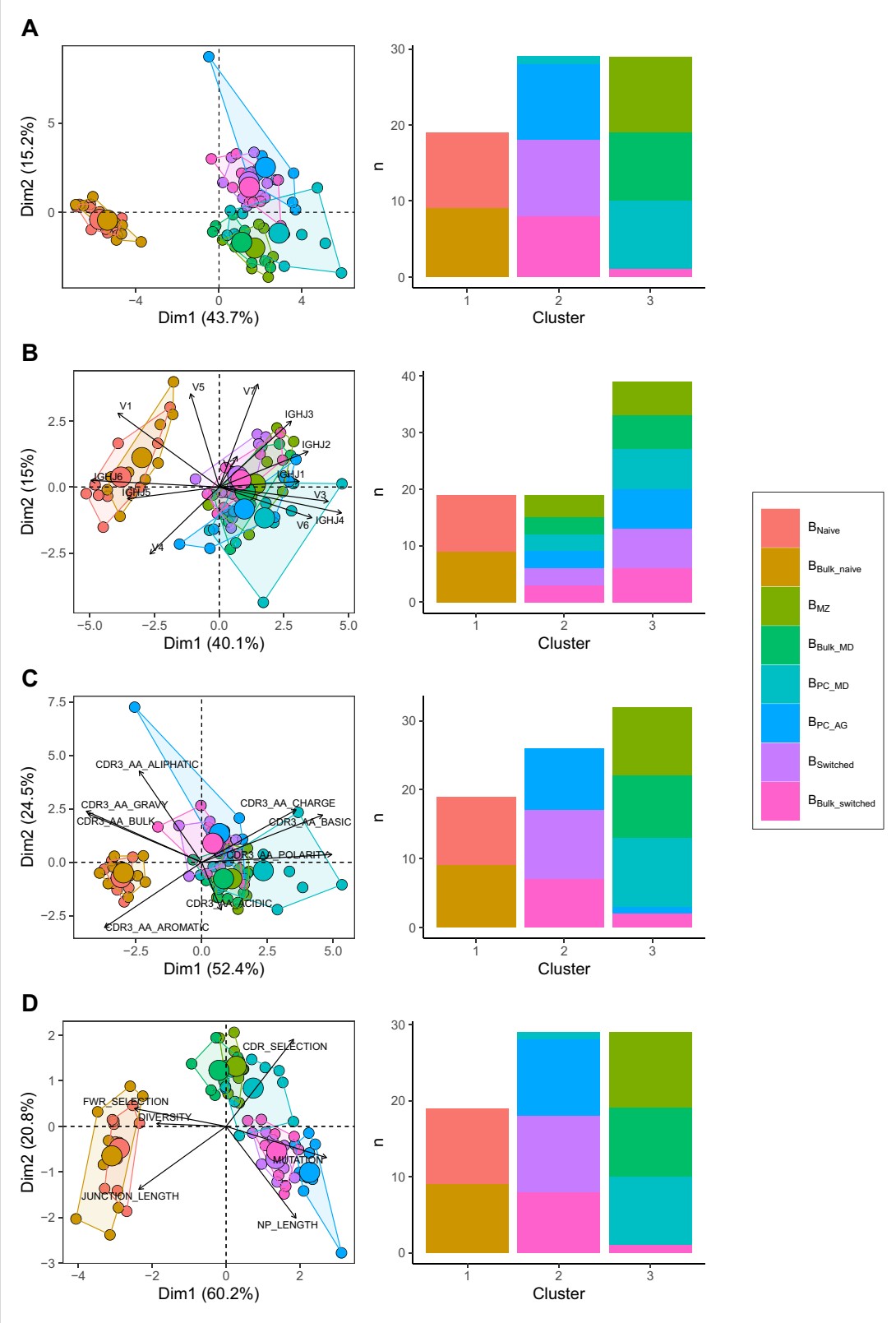

**Figure 1.** Different repertoire characteristics similarly separate between B cells subpopulations. Principal component analysis (PCA) (left) and composition of the clusters formed using k-means clustering with k = 3 (right) applied on (**A**) all repertoire characteristics, (**B**) V family and J gene usage (see *Figure 1—figure supplement 1*), (**C**) physiochemical properties of CDR3 junction (see *Figure 1—figure supplement 2*) and (**D**) global repertoire metrics (see *Figure 1—figure supplement 3*). The percentage of all variation in the data that is explained by PC1 and PC2 is shown on the x and y axis,

*Figure 1 continued on next page*

*Figure 1 continued*

respectively, between brackets. In the PCA plots, areas are the convex hulls of the subsets, and the largest point of one colour represents the centre of that hull.

The online version of this article includes the following figure supplement(s) for figure 1:

**Source data 1.** Related to *Figure 1A-D*.

**Figure supplement 1.** Differences in V and J gene usage across different B cell subpopulations.

**Figure supplement 2.** Comparison of CDR3 amino acid physiochemical properties in different B cell subpopulations.

**Figure supplement 3.** Comparison of global repertoire metrics in different B cell subpopulations.

subpopulations mostly driven by higher mutation counts, NP length and selection pressure in the CDR and lower junction length and diversity in $B_{switched}$ compared to $B_{naive}$ (*Figure 1—figure supplement 3*).

In summary, we found that V family and J gene usage, the physiochemical properties of the CDR3 and global repertoire metrics similarly distinguish between B cell subpopulations. $B_{naive}$, $B_{MZ}/B_{PC\_MD}$ and $B_{switched}/B_{PC\_AG}$ were divergent but shared properties with their relative corresponding subsets in the bulk.

## Accurate prediction of cell type based on repertoire features on a single-cell level

We constructed a sequence classifier that predicts the cell type of a sequence using sequence attributes and different repertoire metrics. Since we subsampled our data making our datasets perfectly balanced, we used only accuracy as a performance metric. Logistic regression, decision tree and random forest classifiers all performed satisfactorily (*Figure 2A*). However, logistic regression performed poorly on correctly classifying $B_{switched}$ and $B_{PC\_AG}$, for which accuracy was almost equal to chance. The performance of all three classifiers was highest in distinguishing between $B_{naive}$ and other cell types.

The random forest classifier was the most successful compared to the other two and the most accurate in predicting the cell type of a sequence. We assessed the relevance of specific predictors in properly classifying cell types by calculating feature importance scores for each cell pair (*Figure 2B*). The number of mutations was the highest scoring feature for all cell pairs except for distinguishing between $B_{switched}$ and $B_{PC\_AG}$ and between $B_{MZ}$ and $B_{PC\_MD}$ for which CDR3 amino acid characteristics had higher scores. Within the CDR3 physiochemical properties, average bulkiness, average polarity and the gravy hydrophobicity index were the most differentiating between cell types, whereas the basic and acidic content of the CDR3 chain seemed to be less important. R/S ratio in CDR and FWR and the junction length appeared to have similar scores and were more important in cases where $B_{naive}$ were not one of the two cell types. V family and J gene appeared to have low importance in distinguishing between all cell pairs.

## Within class-switched subsets, sequences with same constant region from different cell types show similar features

When comparing class-switched transcripts originating from $B_{bulk\_switched}$, $B_{switched}$ and $B_{PC\_AG}$, isotype subclasses were similarly distributed. IgA1 was the dominant subclass in IgA transcripts, whereas IgA2 was less frequently used. All cell subpopulations showed a dominant use of IgG1 and IgG2 with little IgG3 and negligible IgG4 (*Figure 3A*). Usage of IgA1 in $B_{PC\_AG}$ was similar to $B_{switched}$ and $B_{bulk\_switched}$ (p=0.28 and p=0.25, Kruskal–Wallis). IgG3 usage was significantly lower in $B_{PC\_AG}$ compared to $B_{bulk\_switched}$ and $B_{switched}$ (p=0.01, p=0.01, Kruskal–Wallis) while IgG1 usage tended to be lower (p=0.13 and p=0.11, Kruskal–Wallis) and IgG2 usage higher in $B_{PC\_AG}$ compared to the other two B cell subpopulations (p=0.11 and p=0.11, Kruskal–Wallis).

When combining repertoire characteristics by isotype subclass and cell type for class-switched transcripts resulting from $B_{bulk\_switched}$, $B_{switched}$ and $B_{PC\_AG}$, we found that samples with the same constant region originating from different cell types overlapped (*Figure 3B*). We identified two clusters: one mainly composed of IgG1 and IgG3 samples from all cell types and another with IgA1, IgA2 and IgG2 samples by applying k-means clustering with k = 2 (*Figure 3C*). By further dividing the data and with increasing k, we observed that newly formed clusters were mainly composed of distinct isotype

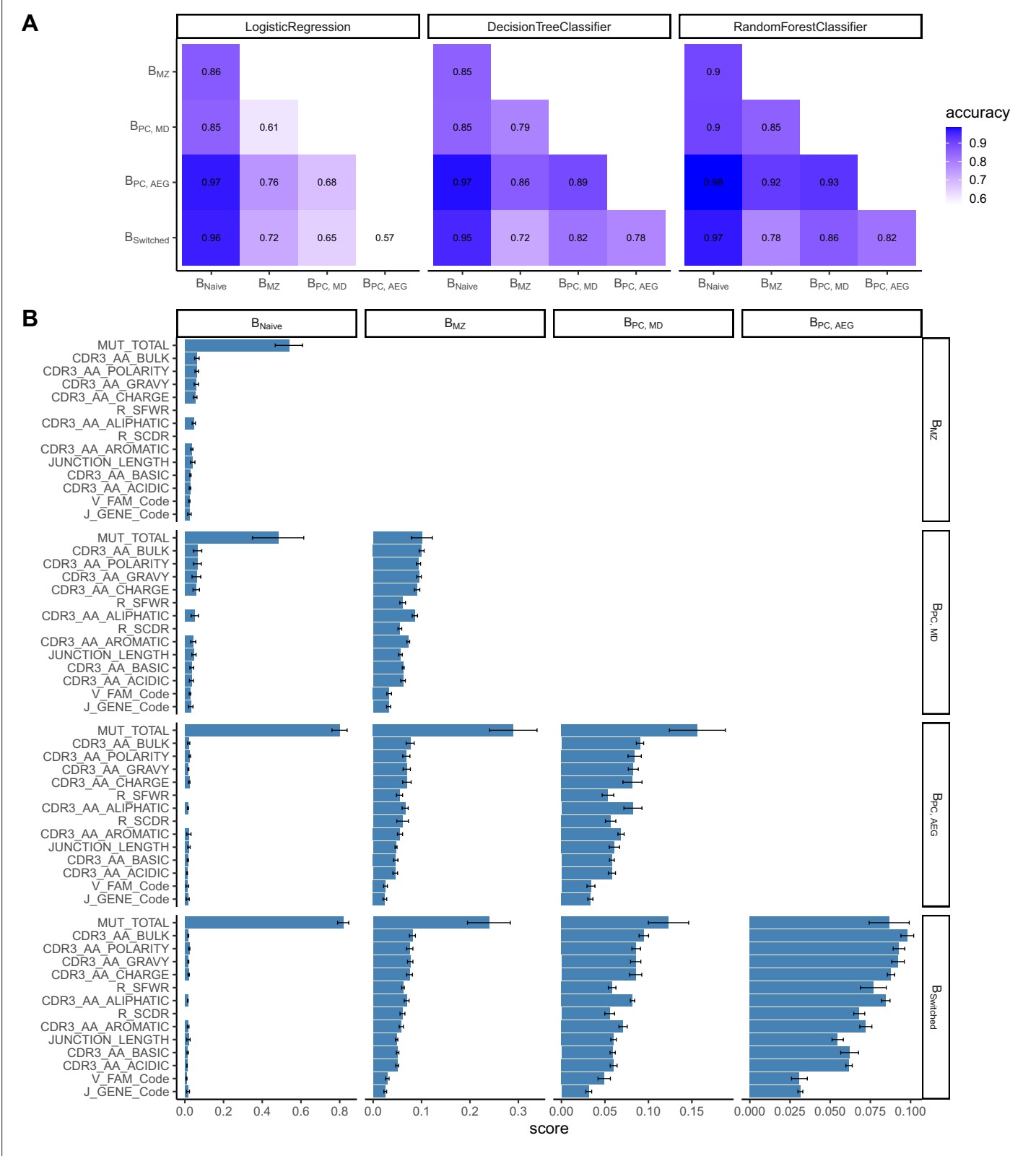

**Figure 2.** Classification accuracies and feature scores on a single-sequence level. (**A**) Heatmap showing pairwise classification accuracy results using logistic regression, decision tree and random forest classifier. (**B**) Random forest feature scores by cell pair.

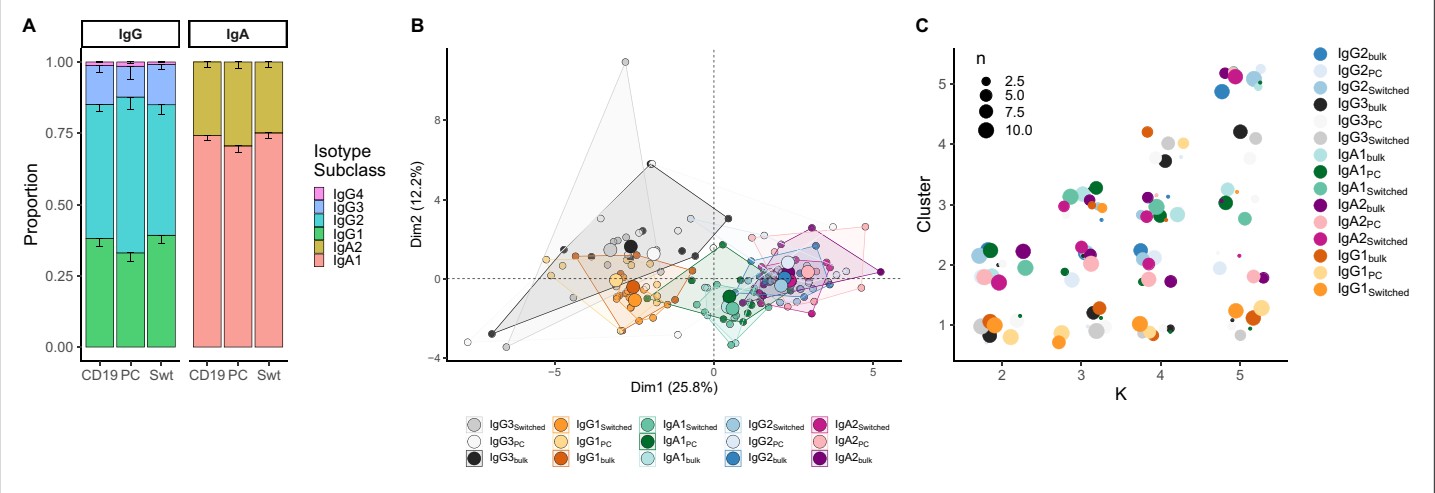

**Figure 3.** Analysis of isotype subclasses in IgG and IgA transcripts. (**A**) Isotype subclass distribution by cell type. Error bars represent the standard error of the mean. (**B**) Principal component analysis (PCA) on all repertoire properties combined by cell type and isotype subclass. Areas are the convex hulls of a group, and the largest point of one colour represents the centre of that hull. (**C**) Composition of the clusters formed by applying the k-means clustering algorithm on all data with increasing k from k = 2 to k = 5. Underlaying source data can be found under *Figure 4—source data 1*.

subclasses, while the cell type itself was not a defining factor for cluster formation. Interestingly, we could not see a clear separation between IgG2 and IgA2 samples with increasing number of clusters.

## B cell repertoire metrics correlate with constant region positioning on the IgH locus in class-switched subsets

The IgH locus contains nine constant genes. The genes encoding for IgM and IgD are the closest to the V-D-J recombination sites while those for IgG3, IgG1 and IgA1 are further downstream but still close to IgM/IgD, whereas more distant on the locus are the genes that encode for IgG2, IgG4, IgE and IgA2 (*Figure 4A*). We determined and compared B cell repertoire metrics between different subclasses in $B_{PC}$ and $B_{switched}$ and compared those to $B_{naive}$ and $B_{MZ}$. $B_{naive}$ showed the lowest number of mutations and R/S ratio and longest CDR3 junction. Memory subsets had a high number of mutations, with $B_{MZ}$ and $B_{PC\_MD}$ having fewer mutations than class-switched transcripts (*Figure 4B*). IgM-distal subclasses IgG2 and IgA2 in both $B_{switched}$ and $B_{PC\_AG}$ showed the highest R/S ratio, indicating high selection pressure (*Figure 4C*). All antigen-experienced subsets had a lower junction length compared to $B_{naive}$ except for IgM-proximal transcripts IgG3 and IgG1 (*Figure 4E*). The proportion of IGHV4-34, the gene associated with self-reactivity (*Bashford-Rogers et al., 2018*), was lower in memory subsets compared to $B_{naive}$ except for IgG3 from $B_{switched}$ for which the proportion of IGHV4-34 was similar to naïve subsets (*Figure 4F*). Within IgG and IgA sequences, genomic distance from IgM correlated with a higher R/S ratio, shorter junction and lower usage of IGHV4-34. $B_{PC}$ had a significantly lower diversity compared to all other cell types (*Figure 4G*). Interestingly, transcripts from $B_{switched}$ showed a similar diversity to $B_{naive}$, whereas $B_{MZ}$ were less diverse. Within $B_{PC\_AG}$, IgM-distal subclasses showed a lower diversity.

IGHV family and IGHJ gene usage also showed a discrepancy between different subsets. IGHV family usage in IgM-proximal subclasses IgG3 and IgG1 was similar to $B_{Naive}$. $B_{MZ}$ and IgM-distal subclasses were enriched in IGHJ4 at the expense of IGHJ6 compared to naïve cells and IgG1-3 B cell subsets (*Figure 4—figure supplement 1*). To reduce the dimensionality of all data points into a single one-dimensional axis, we performed LDA fitted on the relative gene frequencies (*Figure 4H*). This showed a clear distinction between $B_{naive}$, IgG1-3 and $B_{MZ}$, IgG2 and IgA1-2. An LDA fitted on the physiochemical properties of the CDR3 junction also showed a clear distinction between naïve and memory subsets, with IgG3 and IgG1 being closest to $B_{naive}$ and IgG2 and IgA2 overlapping and furthest away (*Figure 4I*).

In summary, we found that in IgA and IgG subclasses different B cell repertoire metrics correlate with the positioning of their respective subclass genes on the IgH locus, namely with the increasing genomic distance from IgM, with the proximal-switched IgH subclasses being more similar to naïve.

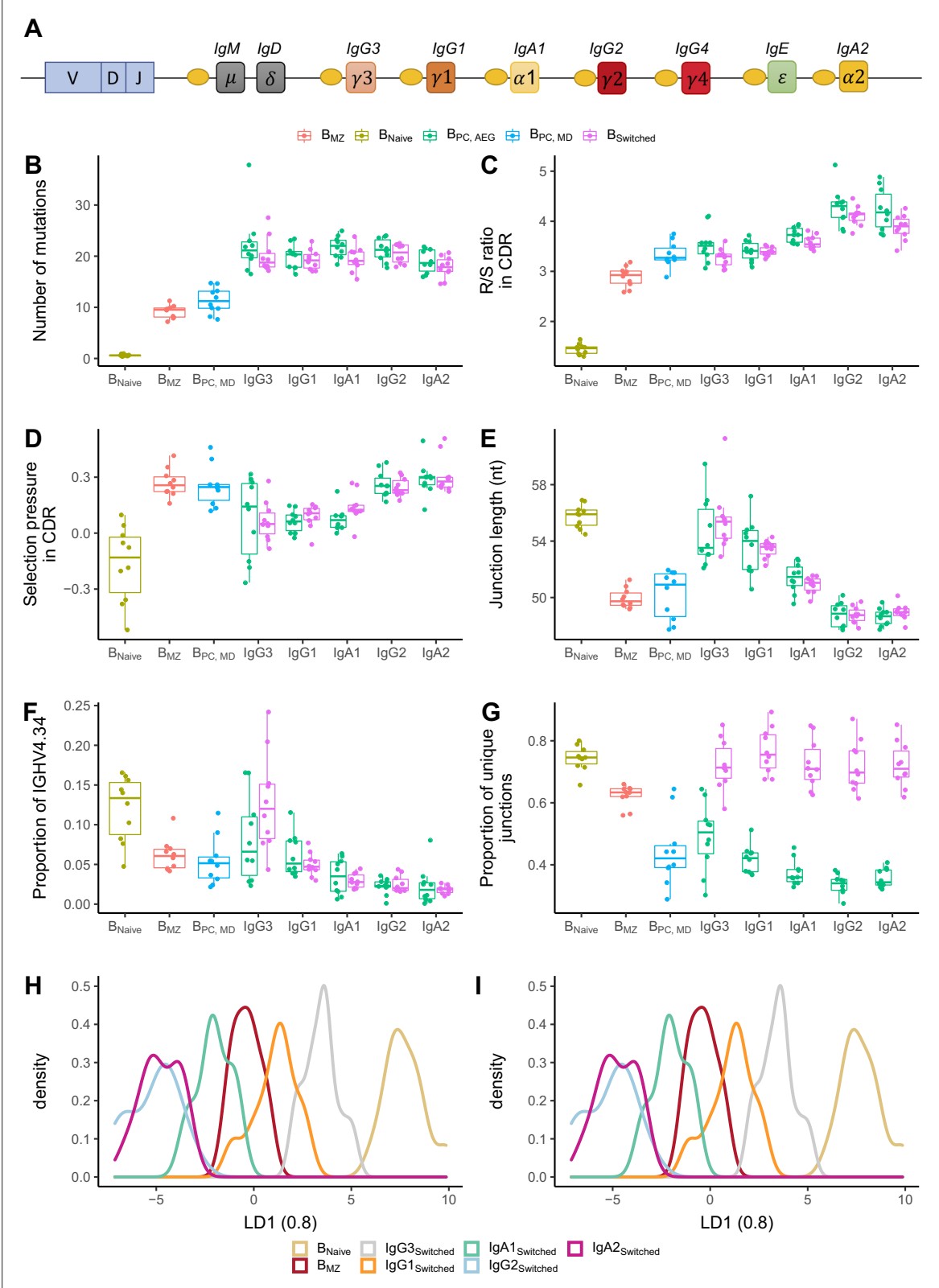

**Figure 4.** Analysis of repertoire metrics by isotype subclass and cell type. (**A**) Overview of the IgH constant region locus. Comparison of (**B**) mutation counts, (**C**) R/S ratio, (**D**) selection pressure, (**E**) junction length, (**F**) proportion of IGHV4.34 and (**G**) diversity between different B cell subpopulations. Linear discriminant analysis (LDA) fitted on (**H**) V family and J gene usage (see *Figure 4—figure supplement 1*) and (**I**) CDR3 amino acid physiochemical properties.

*Figure 4 continued on next page*

*Figure 4 continued*

The online version of this article includes the following figure supplement(s) for figure 4:

**Source data 1.** Related to *Figure 3* and *Figure 4*.

**Figure supplement 1.** Differences in V and J usage across different B cell subpopulations and istotype subclasses.

Memory IgM subsets ($B_{MZ}$ and $B_{PC\_MD}$) were found to have lower mutations and R/S ratio than class-switched subsets. However, in other properties such as junction length, proportion of IGHV4-34 and diversity, they were more similar to IgM-distal subclasses.

## Discussion

Here, we used AIRR-seq to characterise similarities and differences in the IgH repertoire of bulk B cells and different sorted naïve and memory B cell populations. This allowed for an in-depth understanding of the mechanisms underlying B cell responses. We report differences in V family and J gene usage, CDR3 physiochemical properties and global repertoire characteristics that similarly distinguish between naïve, IgM/IgD memory and class-switched subsets both at the repertoire and at the sequence level. Furthermore, we show alterations in the repertoire characteristics at the isotype subclass level unrelated to cell type that correlate with the position of the constant gene on the IgH locus. This study provides powerful insight on biological mechanisms underlying the B cell response as well as novel understanding of AIRR-seq methodologies to be taken into account in future studies.

Earlier studies have revealed differences in gene usage and repertoire properties between different sorted B cell subpopulations (*Mroczek et al., 2014*; *Wu et al., 2010*; *Briney et al., 2012a*; *Larimore et al., 2012*; *DeWitt et al., 2016*; *Bautista et al., 2020*). In addition, some differences in repertoire characteristics across isotype subclasses have been determined by using bulk B cells as starting material (*Jackson et al., 2014*; *Kitaura et al., 2017*; *de Jong et al., 2017*). However, studies combining IgH sequencing from both bulk and sorted B cell subpopulations have been sparse. Here, we sought to compare IgH repertoire properties between transcripts sorted *in silico* from bulk B cells and physically sorted B cell subpopulations. In addition, using isotype-resolved sequencing, we aimed to perform a comparative analysis of the IgH repertoire of transcripts with the same constant region originating from different cell types and correlate the differences in repertoire with position on the IgH locus. The type of material obtained, the sequencing depth and the experimental and computational methods used here constitute by far the most thorough and comprehensive analysis of repertoire differences between B cell subsets and isotype subclasses and thereby provide unique insights into the underlying mechanisms, leading to the formation of different B cell subpopulations.

Previous work involving human naïve and antigen-experienced B cell repertoires havs shown naïve B cells to have shorter junctions and higher usage of IGHJ6 and IGHV3, and lower usage of IGHJ4 and IGHV1 compared with IgM memory and switched B cells (*Briney et al., 2012a*; *Larimore et al., 2012*; *DeWitt et al., 2016*; *Bautista et al., 2020*). Differences in gene usage and CDR3 properties between IgM memory and switched B cells have also been reported (*Wu et al., 2010*). IgM memory and switched B cells have been found to use more negatively charged residues and to have less hydrophobic junctions compared with naïve B cells (*Mroczek et al., 2014*; *Wu et al., 2010*). Here, we focused on a more detailed examination of the repertoires by combining multiple characteristics using dimensionality reduction methods. Results of a previous study revealed that combining only a few repertoire characteristics is sufficient to discriminate between B cell subpopulations (*Galson et al., 2015*). In addition, an LDA combining V gene family proportions has been found to successfully distinguish between IgM and IgG repertoires (*Friedensohn et al., 2018*). We extend these findings by showing that using V family and J gene usage, CDR3 physiochemical properties or global repertoire characteristics similarly allow to separate between naïve and memory subpopulations. This suggests that distinct B cell subpopulations derive from different developmental mechanisms and are subject to selective processes that lead to similar variable gene identity. The importance of the CDR3 and its location at the centre of the antigen-binding site makes the observed differences in physiochemical properties and junction length interesting. These findings may suggest that different B cells subpopulations stimulated by diverse antigens have distinct junction compositions and properties. Further work to investigate CDR3-specific residue content and prediction of CDR3 protein structures could be enriching for antibody specificity and antigen recognition in different B cell subpopulations.

Previous research has demonstrated that same B cell subpopulations from different donors are more similar in their repertoire characteristics than different B cell subpopulations within an individual (*Briney et al., 2012b*; *Rubelt et al., 2016*). This has led to the understanding that differences between naïve and memory cells are conserved across unrelated individuals. Our findings are in agreement with these observations, and we extend on those by showing that the main defining factor in repertoire similarity is the constant region type, namely the isotype subclass, and that differences between subclasses are conserved across both cell type and individual. This finding suggests the existence of an isotype-based mechanism for repertoire control that is constant across cell types and individuals.

In addition to the comparative analysis of the different peripheral B cell subsets, our study represents, to our knowledge, the first comparison of bulk B cell sequencing with sorted B cell subpopulations. We showed that sequencing unsorted B cells from peripheral blood and combining the constant region information with the degree of SHM to bioinformatically group transcripts yields accurate results comparable to physical sorting, especially when analysing global repertoire characteristics. We acknowledge that this might be limiting in tasks sensitive to potential biases from different RNA levels per cell such as identifying antigen-specific sequences from PCs.

Recent IgH repertoire studies have moved towards using machine learning and artificial intelligence in contrast to traditional statistical approaches for goals including vaccine design, immunodiagnostics and antibody discovery (*Greiff et al., 2017*; *Ostmeyer et al., 2017*; *Konishi et al., 2019*; *Shemesh et al., 2021*). Previous work has focused on representing repertoires as sequence or subsequence-based features, that is, overlapping amino acid k-mers and their Atchley biophysiochemical properties (*Greiff et al., 2017*; *Ostmeyer et al., 2017*). Here, we report a simple pairwise classifier that successfully predicts the cell type of a sequence based on only the commonly used sequence attributes such as number of mutations and junction length. Random forest and decision tree classifiers outperformed the logistic regression algorithm suggesting a non-linear separation between cell types. A common concern when applying machine learning is the possibility of over-fitting. To prevent this, we trained the algorithm on 80% of the data and tested its performance on the remaining unseen 20%. We also subsampled every pair of classes to equal number of sequences in order to balance the dataset. The model presented here is applied only within an individual and is thereby confined by repertoire signals that might be individual-specific. More work improving the generalisability of the model across individuals would be revolutionising in terms of its potential practical applications. Unsurprisingly, the number of mutations was the most important feature in distinguishing between most cell types. These results, along with previous work, are promising and suggest that increasing the predictive potential of machine learning methods could help in finding sequence characteristics that distinguish between groups, such as disease state and healthy.

Studies indicate that both direct and sequential CSR to IgM-distal isotype subclasses can occur (*Wesemann et al., 2011*; *Looney et al., 2016*). Several studies have provided evidence for sequential CSR. IgM was found to commonly switch to proximal subclasses (IgG1, IgA1 and IgG2), but direct switches from IgM to more downstream subclasses (IgG4, IgE or IgA2) were rare (*Horns et al., 2016*). It has also been reported that a deficiency in IgG3, the most IgM-proximal subclass, frequently results in a decrease in other IgG subclasses (*Meyts et al., 2006*). Although it is challenging to determine whether sequential CSR occurs during a primary response, by re-entry into the germinal centre or during a secondary response to the same antigen, we and others have shown that IgM-distal subclasses accumulate with age, likely due to secondary encounter with antigen (*Ghraichy et al., 2020*; *IJspeert et al., 2016*). Studies comparing the mean mutation number between isotype subclasses have shown contradicting results. In one study, mutations varied in relation to the constant region position on the IgH locus, with the closest to IgM (IgG3) having the lowest mutations (*Jackson et al., 2014*), while in another study, no such difference was observed (*Kitaura et al., 2017*). We did not find a difference in number of mutations among IgG subclasses. Our findings rather suggest that mutation is more efficient in more downstream subclasses as we found that these exhibit higher R/S ratios and selection pressure in the CDR, consistent with previous studies (*de Jong et al., 2017*). Generally, class-switched IgM-distal subclasses showed signs of maturity (shorter junctions, lower IGHV4-34 usage) while transcripts from class-switched IgM proximal subclasses were more similar to those of naïve B cells. These results suggest that sequential CSR subjects B cells to selective forces, leading to more mature variable gene properties without necessarily accumulating more mutations.

In summary, in this study we took an extensive look at the IgH repertoire of different flow cytometry sorted as well as bioinformatically grouped cell types and isotype subclasses of healthy individuals. Using advanced bioinformatic tools, statistical analysis and machine learning, this analysis provides deep insight into the different mechanisms of B cell development and boosts our understanding of the B cell system components in health. The data and methods presented here provide a foundation for future work investigating the immune repertoire of patients with altered immune status and hold promise for the application of AIRR-seq along with machine learning techniques in clinical and diagnostic settings.

# Materials and methods

## Key resources table

| Reagent type (species) or resource | Designation | Source or reference | Identifiers | Additional information |
|---|---|---|---|---|
| Antibody | Anti-human CD19-PE (mouse monoclonal) | BioLegend | Cat# 363003; RRID:AB_2564125 | FACS (5 µl per test) |
| Antibody | Anti-human CD27-FITC (mouse monoclonal) | BioLegend | Cat# 302806; RRID:AB_314298 | FACS (5 µl per test) |
| Antibody | Anti-human CD38-APC/Fire 750 (mouse monoclonal) | BioLegend | Cat# 356626; RRID:AB_2616713 | FACS (5 µl per test) |
| Antibody | Anti-human IgD-APC (mouse monoclonal) | BioLegend | Cat# 348222; RRID:AB_2561595 | FACS (5 µl per test) |
| Antibody | Anti-human IgM-Pacific Blue (mouse monoclonal) | BioLegend | Cat# 314513; RRID:AB_10574306 | FACS (5 µl per test) |

## Sample collection and cell sorting

This was a descriptive study, hence no formal sample size calculation was performed. Buffy coat samples were obtained from 10 anonymous healthy adults, hence no approval from the local ethics committee was necessary. B cells were first isolated by magnetic cell sorting using the human CD19 MicroBeads (Miltenyi Biotec, San Diego, CA) and the AutoMACS magnetic cell separator. From 9 out of the 10 samples, $3 \times 10^6$ bulk CD19$^+$ B cells ($B_{bulk}$) were lysed and stored at –80°C. The remaining cells were sorted by flow cytometry into four subpopulations using cell surface markers characteristic for naïve ($B_{naive}$), MZ ($B_{MZ}$), PCs ($B_{PC}$) and switched memory B cells ($B_{switched}$). Cells were then lysed and stored at –80°C. Surface markers, demographics, number of cells and purity of each sample are outlined in **Supplementary file 1**.

## RNA extraction and library preparation

RNA extraction was performed on the lysate using the RNeasy Mini Kit (Qiagen, Hilden, Germany). Libraries were prepared as previously described (**Ghraichy et al., 2020**). Briefly, two reverse transcription (RT) reactions were carried out for each RNA sample resulting from $B_{bulk}$ or $B_{PC}$: one with equal concentrations of IgM and IgD-specific primers and another with IgA, IgG and IgE-specific primers. Only one RT reaction with IgM and IgD-specific primers was performed on $B_{naive}$ and $B_{MZ}$ samples; similarly, we applied one RT reaction with IgA, IgG and IgE primers on samples obtained from $B_{switched}$. IgH cDNA rearrangements were then amplified in a two-round multiplex PCR using a mix of IGHV region forward primers and Illumina adapter primers, followed by gel extraction for purification and size selection. The final concentration of PCR products was measured using Qubit prior to library preparation and combined with a total of 12 equally concentrated samples. Final libraries barcoded with individual i7 and i5 adapters were sequenced in each run on the Illumina MiSeq platform (2 × 300 bp protocol).

## Data preprocessing

Preprocessing of raw sequences was carried out using the Immcantation toolkit and as per **Ghraichy et al., 2020**; **Vander Heiden et al., 2014**; **Gupta et al., 2015**. Briefly, samples were demultiplexed based on their Illumina tags. A quality filter was applied, paired reads were joined and then collapsed according to their unique molecular identifier (UMI). Identical reads with different UMI were further

collapsed, resulting in a dataset of unique sequences. VDJ gene assignment was carried out using IgBlast (*Ye et al., 2013*). Isotype subclass annotation was carried out by mapping constant regions to germline sequences using stampy (*Lunter and Goodson, 2011*). The number and type of V gene mutations was determined as the number of mismatches with the germline sequence using the R package shazam (*Gupta et al., 2015*). The R package alakazam was also used to calculate the physicochemical properties of the CDR3 amino acid sequences (*Gupta et al., 2015*). Selection pressure was calculated using BASELINe and the statistical framework used to test for selection was CDR_R/(CDR_$R$ + CDR_S) (*Yaari et al., 2012*).

### *In silico* grouping of sequences

For $B_{bulk}$ samples, we used the constant region information combined with the mutation counts to classify individual sequences into different subsets. IgD and IgM sequences with up to 2 nt mutations across the entire V gene were considered 'unmutated' ($B_{bulk\_naïve}$) to account for remaining PCR and sequencing bias. The remaining mutated IgD and IgM sequences were labelled as IgD/IgM memory ($B_{bulk\_MD}$). All class-switched sequences were defined as antigen-experienced regardless of their V gene mutation count ($B_{bulk\_switched}$). We split the sequences originating from $B_{PC}$ into two categories: IgM/IgD $B_{PC}$ ($B_{PC\_MD}$) and switched IgG/IgA PCs ($B_{PC\_AG}$) according to the constant region of the sequences.

### Summarising repertoire characteristics

V family and J gene usage was calculated in proportions for each individual and cell type. We summarised the mean of the following CDR3 physiochemical characteristics: hydrophobicity, bulkiness, polarity, normalised aliphatic index, normalised net charge, acidic side chain residue content, basic side chain residue content, aromatic side chain content by individual and cell type.

Mean junction length, number of mutations and numbers of non-template (N) and palindromic (P) nucleotide added at the junction were calculated by individual and cell type. Selection pressure was summarised separately in CDR and framework region (FWR). Diversity was calculated as the proportion of unique junctions out of total transcripts. The preceding characteristics are referred to as global repertoire metrics.

### Dimensionality reduction and clustering

PCA and k-means clustering were applied to the different repertoire characteristics to explore and find associations in the data. They were applied using the internal R functions prcomp() and kmeans() (*R Development Core Team, 2018*). LDA was performed using the R function lda() from the package MASS (*Venables and Ripley, 2002*).

### Sequence classifier

We constructed the sequence classifier using the sklearn package in Python (*Pedregosa, 2015*). Because we have the constant region information and to avoid error accumulation, we performed a pairwise classification, thereby transforming the multiclass problem into a binary classification. Within every participant and for every pair of cells, we subsampled to the lower sequence number to avoid bias and dataset imbalance. We used the number of mutations, the physiochemical properties and the junction length as numerical input features. The V gene family and J gene were one-hot encoded. In the case where the naïve cells were not one of the two classes, the replacement/silent (R/S) mutation ratios in CDR and FWR were included as features. We split the data into training and testing set using the default test size of 0.2. We used logistic regression, decision tree and random forest classifiers for prediction. The accuracy was recorded to judge the overall performance of the models. For every pair of classes, the mean accuracy of the 10 samples was calculated.

### Acknowledgements

This work was supported by Swiss National Science Foundation (Ambizione-SCORE. PZ00P3_161147 and PZ00P3_183777) (JT); Gottfried und Julia Bangerter-Rhyner-Stiftung (JT); Olga Mayenfisch Stiftung (JT); Palatin-Stiftung (JT); Investment fund of the University of Zurich (JT). Biotechnology and Biological Sciences Research Council (BBSRC) (AK); UCB Pharma Ltd (AK); Royal Comission for the Exhibtion of 1851 Industrial Fellowship (AK).

# Additional information

## Competing interests

Jacob D Galson: employee of Alchemab Therapeutics Limited but there is no conflict of interest related to this work. Charlotte M Deane: Reviewing editor, *eLife*. The other authors declare that no competing interests exist.

## Funding

| Funder | Grant reference number | Author |
|---|---|---|
| Swiss National Science Foundation | PZ00P3_161147 | Johannes Trück |
| Swiss National Science Foundation | PZ00P3_183777 | Johannes Trück |
| Gottfried und Julia Bangerter-Rhyner-Stiftung | | Johannes Trück |
| Olga Mayenfisch Stiftung | | Johannes Trück |
| Palatin-Stiftung | | Johannes Trück |
| Biotechnology and Biological Sciences Research Council | BB/M011224/1 | Aleksandr Kovaltsuk |
| UCB Pharma Ltd | | Aleksandr Kovaltsuk |
| Royal Commission for the Exhibition of 1851 Industrial Fellowship | | Aleksandr Kovaltsuk |

The funders had no role in study design, data collection and interpretation, or the decision to submit the work for publication.

## Author contributions

Marie Ghraichy, Data curation, Formal analysis, Investigation, Methodology, Resources, Software, Visualization, Writing – original draft, Writing – review and editing; Valentin von Niederhäusern, Data curation, Formal analysis, Methodology, Resources, Writing – review and editing; Aleksandr Kovaltsuk, Conceptualization, Funding acquisition, Methodology, Writing – review and editing; Jacob D Galson, Methodology, Software, Writing – review and editing; Charlotte M Deane, Funding acquisition, Supervision, Writing – review and editing; Johannes Trück, Conceptualization, Funding acquisition, Methodology, Project administration, Supervision, Writing – original draft, Writing – review and editing

## Author ORCIDs

Marie Ghraichy ⬤ http://orcid.org/0000-0002-7348-796X
Jacob D Galson ⬤ http://orcid.org/0000-0003-4916-800X
Charlotte M Deane ⬤ http://orcid.org/0000-0003-1388-2252
Johannes Trück ⬤ http://orcid.org/0000-0002-0418-7381

## Ethics

Human subjects: Buffy coat samples were obtained from 10 anonymous healthy adults, hence no approval from the local ethics committee was necessary.

## Decision letter and Author response

Decision letter https://doi.org/10.7554/73111.sa1
Author response https://doi.org/10.7554/73111.sa2

# Additional files

## Supplementary files

• Supplementary file 1. Demographics, number of cells and purity of samples.
• Transparent reporting form

## Data availability

Raw data used in this study are available at the NCBI Sequencing Read Archive (https://www.ncbi.nlm.nih.gov/sra) under BioProject number PRJNA748239 including metadata meeting MiAIRR standards (32). The processed dataset is available in Zenodo (https://doi.org/10.5281/zenodo.3585046) along with the protocol describing the exact processing steps with the software tools and version numbers.

The following dataset was generated:

| Author(s) | Year | Dataset title | Dataset URL | Database and Identifier |
| --- | --- | --- | --- | --- |
| Ghraichy M, Trück J | 2021 | IgH repertoire sequencing in human B cell subpopulations | https://www.ncbi.nlm.nih.gov/bioproject/PRJNA748239/ | NCBI BioProject, PRJNA748239 |
| Ghraichy M, Trück J | 2019 | Pre-processed B cell receptor repertoire sequencing data from BioProject PRJNA527941 | https://doi.org/10.5281/zenodo.3585046 | Zenodo, 10.5281/zenodo.3585046 |

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
