## [Decision Letter]

**Acceptance summary:**

The described findings are a timely update to fulfill the gaps of previously missed issues to uncover IgH repertoires in different isotype-specific groups and shed significant lights on a better understanding of adaptive immune systems.

**Decision letter after peer review:**

Thank you for submitting your article "Different B cell subpopulations show distinct patterns in their IgH repertoire metrics" for consideration by *eLife*. Your article has been reviewed by 2 peer reviewers, and the evaluation has been overseen by a Reviewing Editor and Betty Diamond as the Senior Editor. The following individual involved in review of your submission has agreed to reveal their identity: Wanli Liu (Reviewer #1).

Essential revisions:

1) As the reviewer 2 has voiced, by fairly referring previous literatures, authors should mention what is the unique outcome from this study, compared with our previous knowledge; indeed, differences in IgH repertoire among different B cell subsets has been already demonstrated by many previous studies. In addition, authors should mention biological implications from this study in a way easily understood by the biologists.

2) Age dependence is important. Thus, analysis of the correlation and proportion of mean mutation concerning the age of enrolled people should be done.

3) The authors demonstrated that IgH of different Ig classes and IgG subclasses show distinct repertoire depending on the distance from Cmu. This finding is novel. However, IgH repertoire of marginal zone B cells and plasma cells producing IgM/IgD supports this conclusion only in some parameters such as number of somatic mutations but not other parameters such as junction length. Moreover, proportion of unique junctions is more influenced by plasma cell differentiation than Ig classes. Therefore, the conclusion of this part needs to be revised.

4) The minimal sampling proportion required to perform the study is of crucial importance. In the material and method section, references should be incorporated, suggesting how the sampling size of 10 healthy individuals has been selected.

5) At the end of the Discussion part, a concluding sentence should be incorporated, illustrating how these algorithmic tools, together with machine learning techniques, can advance our understanding of immunology and address unmet clinical needs related to infectious diseases, immune dysregulation.

*Reviewer #1:*

Brief Summary of achievement: This potentially important study utilizes advanced statistical methods and machine learning algorithms, and bioinformatics approaches to investigate the IgH repertoire of different B cell and subpopulation cell types to determine whether, physically sorted B cell repertoire shares similar characteristics with their corresponding subsets in bulks. In the current study, the authors solved the connection that transcripts from mechanically sorted B cell subgroups have identical qualities to their bulk counterpart's subsets, which were clustered *in silico* using isotype subclass knowledge and the multitude of mutations. The authors further showed that different cell types corresponding to the same isotype subclass have similar sequences, implying that the IgH repertoire contains isotype-specific rather than cell-type-specific characteristics. Most impressively, the authors link these signatures to isotype subclass location on the locus and discover that downstream subclasses have more mature signals. Overall, this is a very impressive and innovative approach for studying the process and adds another layer of information into understanding the mechanisms underlying B-cell responses and uncover critical relevance of AIRR-seq approaches that will be used in future research.

Author's contribution and structure of the proposed study: The prescribed findings would be a timely update to fulfill the gaps of previously missed issues to uncover IgH repertoires in different isotype-specific groups and shed significant light on a better understanding of adaptive immune responses. The Candidate has achieved the proposed issues precisely. The findings are presented clearly and concisely, with adequate explanation and interpretation. The results support the theme logically, and the methods have been implemented in a balanced way to achieve the results. The discussion provides a critical elaboration of the results obtained. These studies in the future may provide detailed insights for adaptive immune researchers struggling to develop therapeutic drugs to treat several autoimmune and BCRs structural interlinked diseases. However, including autoimmune disease patients could further help researchers get detailed insights to extract the precise conclusion.

Strengths and weaknesses of the study: This study provides detailed insights into healthy individuals' IgH repertoire characteristics. Previous investigations have concentrated mainly on showing repertoires as a sequence or subsequence-based features. However, these investigations provided a brief pairwise predictor that effectively predicts a sequence's cell type based on the most often utilized sequence parameters, such as the number of mutations and junction length. On the other hand, these studies provide investigations within individuals. However, the feasibility of the model should be enhanced across the individuals. Moreover, interrogating the importance of IgH repertoire characteristics in B cells subsets during diseased conditions or concerning specific age should be paid explicit attention in the future through applying these sound approaches.

Before the publication, the following few points should be well considered:

1 – The authors use ten healthy individuals as readout; however, how V and J repertoire characteristics behave in people of different ages are missing. Secondly, the mean mutation parameter has been interestingly discussed in these studies with isotype-specific subclasses suggesting that, mutation most commonly is seen in downstream subclasses. As per my opinion, it would be interesting to analyze the correlation and proportion of mean mutation concerning the age of enrolled people. The authors can beautifully select people of different ages to analyze the results.

2 – Few terminologies such as PCA, LDA, CDR3 are used at high frequency; I suggest elaborating their abbreviation in the introduction section for better clarity.

3 – At the end of the Discussion part, a concluding sentence should be incorporated, illustrating how these algorithmic tools, together with machine learning techniques, can advance our understanding of immunology and address unmet clinical needs related to infectious diseases, immune dysregulation.

4 – The minimal sampling proportion required to perform the study is of crucial importance. In the material and method section, references should be incorporated, suggesting how the sampling size of 10 healthy individuals has been selected.

These studies comprehensively provided a model to check IgH repertoire within individuals, specifically in healthy individuals, possibly confined by individual-specific repertoire signals. However, it would be fascinating to incorporate diseased individuals in the study to elaborate on the system's feasibility. I would suggest adding few lines in the discussion part as a dimension of future work.

*Reviewer #2:*

In this paper, the authors analyzed IgH repertoire of various B cell subsets from human donors such as naïve, marginal zone and switched B cells and plasma cells producing IgM or IgD and those producing IgA or IgG using bioinformatic analysis and machine learning. They also analyzed IgH repertoire in bulk and *in silico* separated B cell subsets. The authors demonstrated that marginal zone B cells and plasma cells producing IgM or IgD show similar IgH repertoire whereas IgH repertoire in switched B cells is similar to that of plasma cells producing IgG or IgA but distinct from the other subsets. Number of somatic mutations, physiochemical nature of CDR3 and usage of VH and JH contribute to the distinction of IgH repertoire between different subsets.

The authors comprehensively analyzed IgH repertoire and nicely showed the difference between different B cell subsets. However, difference in IgH repertoire between different B cell subsets has been demonstrated by various previous studies. Therefore, what biological questions can be addressed by this analysis is somewhat unclear.

The difference of physiochemical properties of CDR3 between different subsets is interesting. Could the author discuss the impact of this difference in antigen binding?

The authors demonstrated that IgH of different Ig classes and IgG subclasses show distinct repertoire depending on the distance from Cmu. This finding is novel. However, IgH repertoire of marginal zone B cells and plasma cells producing IgM/IgD supports this conclusion only in some parameters such as number of somatic mutations but not other parameters such as junction length. Moreover, proportion of unique junctions is more influenced by plasma cell differentiation than Ig classes. Therefore, the conclusion of this part needs to be revised.

---

## [Author Response]

Essential revisions:1) As the reviewer 2 has voiced, by fairly referring previous literatures, authors should mention what is the unique outcome from this study, compared with our previous knowledge; indeed, differences in IgH repertoire among different B cell subsets has been already demonstrated by many previous studies. In addition, authors should mention biological implications from this study in a way easily understood by the biologists.

We have added the following paragraph to our discussion clearly stating our research questions and novelty in contrast with previous work:

“Earlier studies have revealed differences in gene usage and repertoire properties between different sorted B cell subpopulations. In addition, some differences in repertoire characteristics across isotype subclasses have been determined by using bulk B cells as starting material. However, studies combining IgH sequencing from both bulk and sorted B cell subpopulations have been sparse. Here, we sought to compare IgH repertoire properties between transcripts sorted *in silico* from bulk B cells and physically sorted B cell subpopulations. In addition, using isotype-resolved sequencing, we aimed to perform a comparative analysis of the IgH repertoire of transcripts with the same constant region originating from different cell types and correlate the differences in repertoire with position on the IgH locus. The type of material obtained, the sequencing depth, and the experimental and computational methods used here constitute by far the most thorough and comprehensive analysis of repertoire differences between B cell subsets and isotype subclasses and thereby provide unique insights into the underlying mechanisms leading to formation of different B cell subpopulations.”

2) Age dependence is important. Thus, analysis of the correlation and proportion of mean mutation concerning the age of enrolled people should be done.

We agree with the reviewer that age is an important factor in shaping the IgH repertoire. We and others have previously looked at the effect of age on the IgH repertoire characteristics in detail (Ghraichy, 2020; IJspeert, 2016). These studies have shown that maturation of the B cell compartment with age is primarily observed as an increase in the number of mutations, with most changes occurring in the first decade of life. Changes continue to be made thereafter, but at a much lower rate. Using the data in the present manuscript, additional correlation analysis was conducted for all repertoire characteristics that have been shown to be age dependent in a previous analysis using the same sequencing and analysis protocol (Ghraichy, 2020). This analysis revealed a positive correlation between the number of mutations and age, and between R/S ratio in CDR and age but only in IgG2 transcripts. No age-dependent correlation was observed for any other subsets. However, because of the number of individuals included and their age range, we decided not to include this analysis as we thought it did not add any important value to previous findings. Because our proposed results are based on an intraindividual and cross-subgroup analysis, we believe that our results are valid in general, despite isolated age dependence.

3) The authors demonstrated that IgH of different Ig classes and IgG subclasses show distinct repertoire depending on the distance from Cmu. This finding is novel. However, IgH repertoire of marginal zone B cells and plasma cells producing IgM/IgD supports this conclusion only in some parameters such as number of somatic mutations but not other parameters such as junction length. Moreover, proportion of unique junctions is more influenced by plasma cell differentiation than Ig classes. Therefore, the conclusion of this part needs to be revised.

We thank the reviewer for this comment and have modified the title and conclusion of this section to specify that parameters are dependent on the distance from Cmu in class-switched subsets. We have also added the following sentence at the end of this section:

“Memory IgM subsets (B_MZ_ and B_PC_MD_) were found to have lower mutations and R/S ratio than class-switched subsets. However, in other properties such as junction length, proportion of IGHV4-34 and diversity, they were more similar to IgM-distal subclasses.”

4) The minimal sampling proportion required to perform the study is of crucial importance. In the material and method section, references should be incorporated, suggesting how the sampling size of 10 healthy individuals has been selected.

Because this is a descriptive study, no formal sample size calculation was performed and the sampling size of 10 was based on the availability of samples. For clarity, we have added a sentence in the material and methods section.

Although additional samples might have been beneficial in increasing the sensitivity for certain analysis, the current approach with full-length isotype-resolved sequencing of 9 bulk and 4 x 10 B cell subpopulations (total n = 49 samples) allowed clear separation of subsets with limited variability and demonstration of overlap between *in silico* and physically sorted B cell subpopulations. Where appropriate (e.g., in machine learning), we used subsampling and divided into a training set and a test set to limit variability in sequencing methods and make data analysis more robust.

5) At the end of the Discussion part, a concluding sentence should be incorporated, illustrating how these algorithmic tools, together with machine learning techniques, can advance our understanding of immunology and address unmet clinical needs related to infectious diseases, immune dysregulation.

According to the reviewer’s suggestion, we have added the following concluding sentence at the end of the discussion:

“The data and methods presented here provide a foundation for future work investigating the immune repertoire of patients with altered immune status and hold promise for the application of AIRR-seq along with machine learning techniques in clinical and diagnostic settings.”

Reviewer #1:Before the publication, the following few points should be well considered :1 – The authors use ten healthy individuals as readout; however, how V and J repertoire characteristics behave in people of different ages are missing. Secondly, the mean mutation parameter has been interestingly discussed in these studies with isotype-specific subclasses suggesting that, mutation most commonly is seen in downstream subclasses. As per my opinion, it would be interesting to analyze the correlation and proportion of mean mutation concerning the age of enrolled people. The authors can beautifully select people of different ages to analyze the results.

We agree with the reviewer that age is an important factor in shaping the IgH repertoire. We and others have previously looked at the effect of age on the IgH repertoire characteristics in detail (Ghraichy, 2020; IJspeert, 2016). These studies have shown that maturation of the B cell compartment with age is primarily observed as an increase in the number of mutations, with most changes occurring in the first decade of life. Changes continue to be made thereafter, but at a much lower rate. Using the data in the present manuscript, additional correlation analysis was conducted for all repertoire characteristics that have been shown to be age dependent in a previous analysis using the same sequencing and analysis protocol (Ghraichy, 2020). This analysis revealed a positive correlation between the number of mutations and age, and between R/S ratio in CDR and age but only in IgG2 transcripts. No age-dependent correlation was observed for any other subsets. However, because of the number of individuals included and their age range, we decided not to include this analysis as we thought it did not add any important value to previous findings. Because our proposed results are based on an intraindividual and cross-subgroup analysis, we believe that our results are valid in general, despite isolated age dependence.

2 – Few terminologies such as PCA, LDA, CDR3 are used at high frequency; I suggest elaborating their abbreviation in the introduction section for better clarity.

We have elaborated the abbreviations in the introduction as suggested.

3 – At the end of the Discussion part, a concluding sentence should be incorporated, illustrating how these algorithmic tools, together with machine learning techniques, can advance our understanding of immunology and address unmet clinical needs related to infectious diseases, immune dysregulation.

According to the reviewer’s suggestion, we have added the following concluding sentence at the end of the discussion:

“The data and methods presented here provide a foundation for future work investigating the immune repertoire of patients with altered immune status and hold promise for the application of AIRR-seq along with machine learning techniques in clinical and diagnostic settings.”

4 – The minimal sampling proportion required to perform the study is of crucial importance. In the material and method section, references should be incorporated, suggesting how the sampling size of 10 healthy individuals has been selected.

We thank the reviewer for this comment. Because this is a descriptive study, no formal sample size calculation was performed and the sampling size of 10 was based on the availability of samples. For clarity, we have added a sentence in the material and methods section.

Although additional samples might have been beneficial in increasing the sensitivity for certain analysis, the current approach with full-length isotype-resolved sequencing of 9 bulk and 4 x 10 B cell subpopulations (total n = 49 samples) allowed clear separation of subsets with limited variability and demonstration of overlap between *in silico* and physically sorted B cell subpopulations. Where appropriate (e.g., in machine learning), we used subsampling and divided into a training set and a test set to limit variability in sequencing methods and make data analysis more robust.

These studies comprehensively provided a model to check IgH repertoire within individuals, specifically in healthy individuals, possibly confined by individual-specific repertoire signals. However, it would be fascinating to incorporate diseased individuals in the study to elaborate on the system's feasibility. I would suggest adding few lines in the discussion part as a dimension of future work.

We agree with the reviewer that applying the methods presented on diseased individuals would be enlightening and have included a sentence at the end of the discussion suggesting this as a possible future direction.

Reviewer #2:In this paper, the authors analyzed IgH repertoire of various B cell subsets from human donors such as naïve, marginal zone and switched B cells and plasma cells producing IgM or IgD and those producing IgA or IgG using bioinformatic analysis and machine learning. They also analyzed IgH repertoire in bulk and *in silico* separated B cell subsets. The authors comprehensively analyzed IgH repertoire and nicely showed the difference between different B cell subsets. However, the novelty of this study is somewhat unclear, and some of the conclusions are not well supported by the data.In this paper, the authors analyzed IgH repertoire of various B cell subsets from human donors such as naïve, marginal zone and switched B cells and plasma cells producing IgM or IgD and those producing IgA or IgG using bioinformatic analysis and machine learning. They also analyzed IgH repertoire in bulk and *in silico* separated B cell subsets. The authors demonstrated that marginal zone B cells and plasma cells producing IgM or IgD show similar IgH repertoire whereas IgH repertoire in switched B cells is similar to that of plasma cells producing IgG or IgA but distinct from the other subsets. Number of somatic mutations, physiochemical nature of CDR3 and usage of VH and JH contribute to the distinction of IgH repertoire between different subsets.The authors comprehensively analyzed IgH repertoire and nicely showed the difference between different B cell subsets. However, difference in IgH repertoire between different B cell subsets has been demonstrated by various previous studies. Therefore, what biological questions can be addressed by this analysis is somewhat unclear.

We have added the following paragraph to our discussion clearly stating our research question and novelty in contrast with previous work:

“Earlier studies have revealed differences in gene usage and repertoire properties between different sorted B cell subpopulations. In addition, some differences in repertoire characteristics across isotype subclasses have been determined by using bulk B cells as starting material. However, studies combining IgH sequencing from both bulk and sorted B cell subpopulations have been sparse. Here, we sought to compare IgH repertoire properties between transcripts sorted *in silico* from bulk B cells and physically sorted B cell subpopulations. In addition, using isotype-resolved sequencing, we aimed to perform a comparative analysis of the IgH repertoire of transcripts with the same constant region originating from different cell types and correlate the differences in repertoire with position on the IgH locus. The type of material obtained, the sequencing depth, and the experimental and computational methods used here constitute by far the most thorough and comprehensive analysis of repertoire differences between B cell subsets and isotype subclasses and thereby provide unique insights into the underlying mechanisms leading to formation of different B cell subpopulations.”

The difference of physiochemical properties of CDR3 between different subsets is interesting. Could the author discuss the impact of this difference in antigen binding?

We thank the reviewer for their comment. To our knowledge, except for charge and hydrophobicity that are associated with self-reactivity, there is limited knowledge how other CDR3 physiochemical characteristics affect antigen binding. However, we have added the following sentence in the discussion suggesting possible future directions to further investigate these changes.

“The importance of the CDR3 and its location at the center of the antigen-binding site make the observed differences in physiochemical properties and junction length interesting. These findings may suggest that different B cells subpopulations stimulated by diverse antigens have distinct junction compositions and properties. Further work to investigate CDR3-specific residue content and prediction of CDR3 protein structures could be enriching for antibody specificity and antigen recognition in different B cell subpopulations.”

The authors demonstrated that IgH of different Ig classes and IgG subclasses show distinct repertoire depending on the distance from Cmu. This finding is novel. However, IgH repertoire of marginal zone B cells and plasma cells producing IgM/IgD supports this conclusion only in some parameters such as number of somatic mutations but not other parameters such as junction length. Moreover, proportion of unique junctions is more influenced by plasma cell differentiation than Ig classes. Therefore, the conclusion of this part needs to be revised.

This is a good point and we have modified the title and conclusion of this section to specify that parameters are dependent on the distance from Cmu in class-switched subsets. We have also added the following sentence at the end of this section:

“Memory IgM subsets (B_MZ_ and B_PC_MD_) were found to have lower mutations and R/S ratio than class-switched subsets. However, in other properties such as junction length, proportion of IGHV4-34 and diversity, they were more similar to IgM-distal subclasses.”